



# Surrogate based aeroelastic design optimization of tip extensions on a modern 10MW wind turbine

Thanasis Barlas, Néstor Ramos-García, Georg Raimund Pirrung, and Sergio González Horcas

DTU Wind Energy, Frederiksborgvej 399, 4000 Roskilde, Denmark

**Correspondence:** Thanasis Barlas (tkba@dtu.dk)

**Abstract.** Advanced aeroelastically optimized tip extensions are among rotor innovation concepts which could contribute to higher performance and lower cost of wind turbines. A novel design optimization framework for wind turbine blade tip extensions, based on surrogate aeroelastic modeling is presented. An academic wind turbine is modelled in an aeroelastic code equipped with a near wake aerodynamic module, and tip extensions with complex shapes are parametrized using 11 design
variables. The design space is explored via full aeroelastic simulations in extreme turbulence and a surrogate model is fitted to the data. Direct optimization is performed based on the surrogate model, seeking to maximize the power of the retrofitted turbine within the ultimate load constraints. The presented optimized design achieves a load neutral gain of up to 6% in annual energy production. Its performance is further evaluated in detail by means of the near wake model used for the generation of the surrogate model, and compared with a higher fidelity aerodynamic module comprising a hybrid filament-particle-mesh
vortex method with a lifting-line implementation. A good agreement between the solvers is obtained at low turbulence levels, while differences in predicted power and flap-wise blade root bending moment grow with increasing turbulence intensity.

## 1 Introduction

The trend of reducing the Levelized Cost of Energy (LCOE) of horizontal axis wind turbines through increasing rotor size has
long been established. To achieve this, the challenges of scale must be overcome through innovative turbine design and control strategies (Veers, 2019). One promising blade design concept is advanced aeroelastically optimized blade tip extensions, which could drive rotor upscaling in a modular and cost effective way.

Existing bibliography relevant to wind turbine applications typically focuses on winglets and aerodynamic tip shapes, purely from an aerodynamics point of view (Johansen, 2006; Gaunaa, 2007; Ferrer, 2007; Chattot, 2009; Elfarra, 2014; Farhan, 2019;
Matheswaran, 2019). Exceptions to this general trend are the recent articles (Zahle, 2018; Sessarego, 2018; Hansen, 2018; Rosemeier, 2020; Horcas, 2020), that put the focus on general blade tip designs and aeroelastic performance. Moreover, there is no relevant research work focusing on performance and design loads of rotors with tip extensions relevant to real operational cases, with a view towards a business case.





In this work, the tip extensions are designed with the objective of maximizing Annual Energy Production (AEP) gain, within
the existing operational load constraints. The relevant business case is associated with improving performance of existing rotors
or customizing rotors for different site conditions, while investing less in new full blade production costs. Due to the fact that
full time domain aeroelastic simulations are utilized for the power and loads evaluation, a surrogate based optimization (SBO)
approach is pursued, in order to avoid issues with gradient evaluations which normally require simplification of the evaluation
cases. Furthermore, the parametrization of the tip extension is detailed enough to represent a blade design optimization ap-
proach, now in a modular way focusing only on the tip. This includes the capability of producing complex shapes with large
sweep and prebend, typically not used in a traditional blade design.

## 2 Aeroelastic model setup

A time domain aeroelastic model of the onshore version of the IEA 10MW RWT (Bortolotti, 2019) was first built. The IEA
10MW RWT is an academic wind turbine model which is the result of aeroelastic optimization of the DTU 10MW rotor. In
particular the DTU 10MW rotor was stretched in order to achieve the maximum AEP gain while satisfying the imposed design
loads constraints. The IEA 10MW RWT design is considered a good representative reference of an optimized modern offshore
wind turbine, where the tip extensions could have a significant impact in the reduction of the LCOE.

A surrogate based optimization framework was then wrapped around the baseline model of the IEA 10MW RWT, with pre-
and post-processing scripts providing the capability of executing simulations of specific tip extensions on the baseline turbine,
with their design variables determined by the optimization routines.

The following sections provide further details of the different components involved in the aforedescribed workflow.

### 2.1 Baseline model

The aeroelastic simulations performed in the present work relied on the commercial software HAWC2 (Larsen, 2007). HAWC2
includes advanced features in the near wake aerodynamic module implementation, providing the ability to accurately simulate
complex tip shapes (Pirrung, 2016, 2017; Li, 2018; Madsen, 2020). In addition to the modified near wake model to account
for blade sweep, the coupled aerodynamic module in HAWC2 uses a non-planar vortex cylinder model (Branlard , 2017) to
compute the effects of prebend and out-of-plane deformation on axial and radial induced velocity.

The in-house multi-fidelity vortex solver MIRAS, [Ramos (2016, 2017)], has been used for a higher fidelity evaluation of the
baseline and the optimized designs. In the present study the lifting line (LL) aerodynamic model is used in combination with
a hybrid filament-particle-mesh flow model (Ramos, 2019). The flow is governed by the vorticity equation, which is obtained
by taking the curl of the Navier-Stokes equation, and describes the evolution of the vorticity of a fluid particle as it moves with
the flow. The coupling between MIRAS and HAWC2 (Ramos, 2020), permits to account for the flexibility of the wind turbine,
as well as the consideration of the effect of the controller and the hydrodynamic loads. MIRAS has been recently modified to
accurately account for blade curvature effects (Li, 2020).





**Table 1.** Design variables and their range.

| variable | length [%] | chord 1 [%] | chord 2 [%] | twist 1 [deg] | twist 2 [deg] | dihedral [deg] | sweep [deg] | sc off [%] | E/G scaling [%] | K opt scaling [%] | pitch opt off [deg] |
|---|---|---|---|---|---|---|---|---|---|---|---|
| min | 5 | 20 | 50 | -10 | -5 | -30 | 0 | 0 | 50 | 100 | -3 |
| max | 7 | 50 | 100 | 5 | 5 | 0 | 30 | 20 | 100 | 150 | 3 |

The power performance and ultimate loads of every design are evaluated in a single load case, comprising an IEC specific DLC1.3 (IEC, 2005) simulation at 8m/s. This case is considered representative for determining the average power performance in below rated power operation and the range of peak loading, since the turbine operation ranges from low power production to full rated power within the simulation time.

## 2.2    Tip extension parametrization

The definition of the tip extension design variables and their design space is probably the most important step in the described optimization process. The variables have been chosen in a way which enables a general blade stretching design capability. Their range is a result of many prior parametric studies, and it is limited to ensure the validity of the aerodynamic modeling. The 11 chosen variables and their extend in the design space are shown in Table 1, with all definitions being relative to the baseline blade. When adding the extensions, the blade is cut at the connection point at 97.5% of its original projected length in

the spanwise coordinate, and the length of the extension is added. The tip extension planform is defined by the chord values at the new tip (chord 1) and the baseline tip (chord 2) positions, and by the twist values at the new tip (twist 1) and the baseline tip (twist 2) positions, all relative to the values at the 97.5% connection point. The relative thickness is defined by assuming the value at the new tip position is equal to the the one at the baseline tip. The distribution of the planform variables is calculated with a cubic Hermite interpolating polynomial. The planforms of two reference tip extensions at the borders of the design space

are compared to the baseline one in Fig. 1. The same distribution approach is utilized for the angle of the section reference line, where a new tip in-plane (sweep) and out-of-plane (dihedral) angles are defined, relative to the existing direction at the connection point. Only backwards sweep and upwind offsets are modelled, since there is no evident design benefit for forward sweep, and upwind dihedral results in a smooth shape continuation of the existing prebend, further away from the tower. The offsets of two reference tip extensions at the borders of the design space are compared to the baseline one in Fig. 2, for a case

of 30 deg sweep and 30 deg dihedral angles. The structural properties of the tip sections are calculated by scaling with the new chord values. Two variables are defined in order to vary the structural characteristics of the tip, accounting for moving the shear center position fore of the baseline position relative to the local chord, and scaling of the flapwise, edgewise and torsional stiffness. The mass and flapwise stiffness of two reference tip extensions at the borders of the design space are compared to the baseline one in Fig. 3. The main controller parameters changing with the tip addition account for the response in below rated

operation, by scaling the rpm-generator torque quadratic gain (K opt), and varying the fine pitch setting (pitch opt).

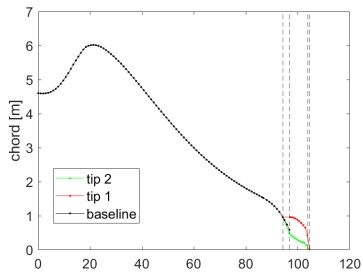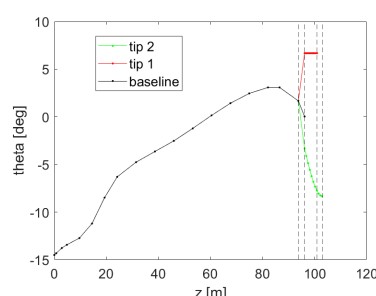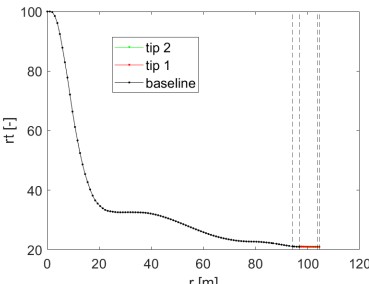

**Figure 1.** Planform for two reference tips at the borders of the design space. Vertical dashed lines indicate the location of the control points.

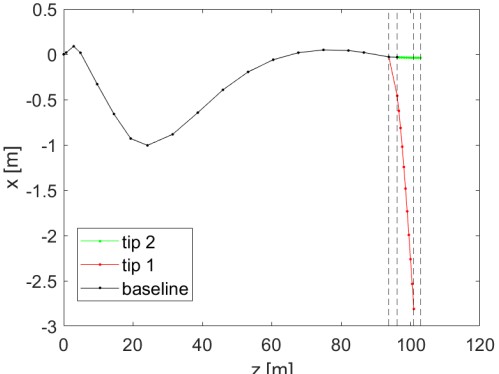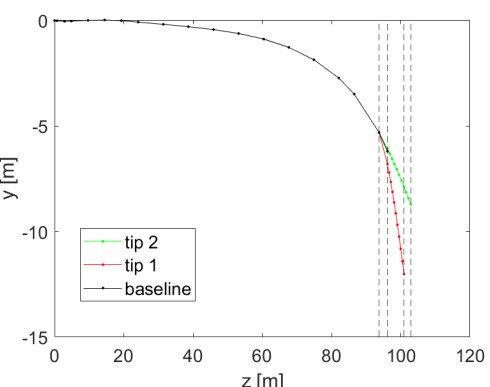

**Figure 2.** Blade centerline for two reference tips at the borders of the design space. Vertical dashed lines indicate the location of the control points.

## 2.3 Pre/post-processing

For every design evaluation loop, the HAWC2 case files are pre-processed, executed and post-processed on a single CPU. The top-level process is shown in Fig. 4. In the pre-processing MATLAB script, the baseline HAWC2 input files are modified in order to generate each tip extension design case. In the HAWC2 model, 10 additional structural and aerodynamic sections are added on the new part of the blade beyond 100% and the sections between 97.5%-100% are modified. The rpm-generator torque quadratic controller gain and the fine-pitch setting are also modified. A case folder with all the HAWC2 input is assembled from all necessary files.

In the post-processing MATLAB script, the output time series files of HAWC2 are processed, and performance statistics are extracted. For the optimization, the mean generator power and the ultimate blade root flapwise bending moment are extracted. For detailed evaluation purposes of the designs, all other component load statistics, and blade distributed outputs are also extracted.



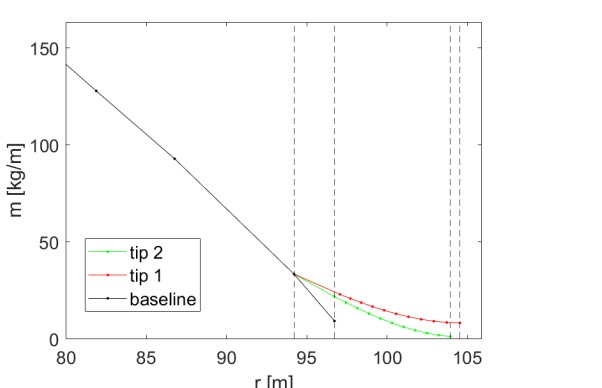 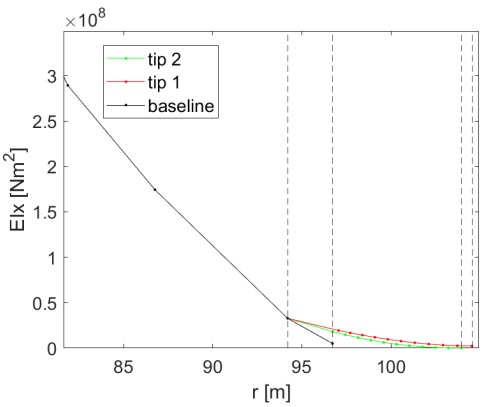

**Figure 3.** Mass and flapwise stiffness for two reference tips at the borders of the design space. Vertical dashed lines indicate the location of the control points.

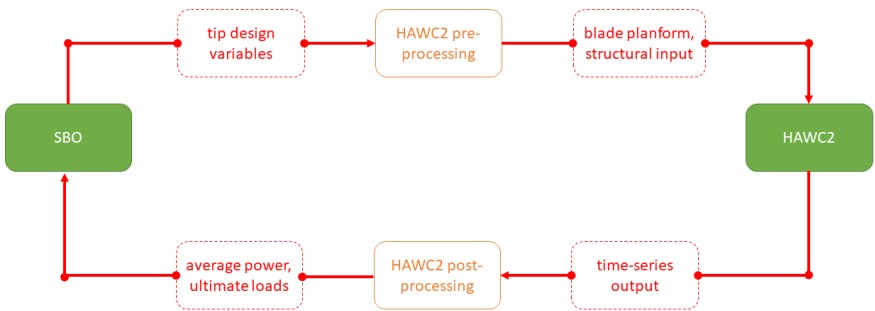

**Figure 4.** SBO setup top-level diagram.

## 3 Surrogate based optimization setup

The SBO framework is setup based on the MATLAB code package MATSuMoTo (Müller, 2013, 2014), which is the MATLAB Surrogate Model Toolbox for deterministic computationally expensive black-box global optimization problems with continuous, integer, or mixed-integer variables that are formulated as minimization problems. The SBO framework determines the design variable sets and send them to the pre-processor to execute the HAWC2 cases, in parallel CPU processing. The general SBO algorithm works as follows:

– Generate initial design sets

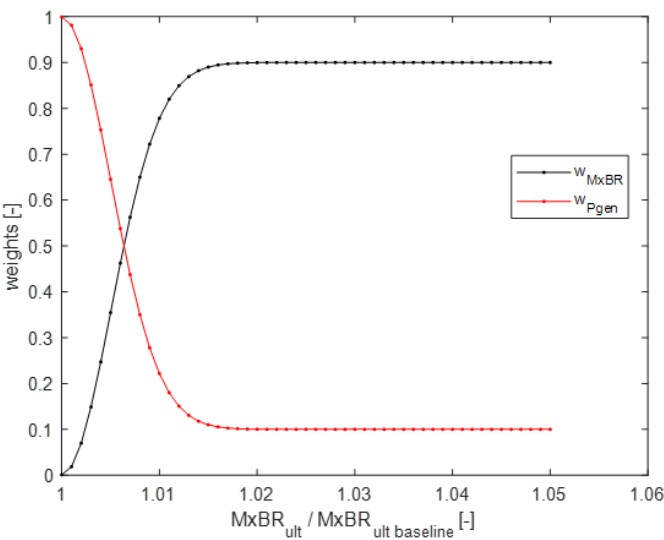

**Figure 5.** Power and load objective function weights as a function of load increase.

- – Do the costly function evaluations at the points generated in the previous step

– Fit a surrogate model to the data

- – Use the surrogate model to predict the objective function values at unsampled points in the variable domain to decide at which points to do the next expensive function evaluations

- – Do the expensive function evaluations at the points selected in the previous step

- – Check if the stopping criterion has been reached. If not, go back to the third step. If the stopping criterion has been met,
stop

The objective function is a very important part of this study, since it determines, which direction in the design space the SBO takes by evaluating new design variable sets. The objective function is defined as a weighted sum of the mean generator power and the ultimate blade root flapwise bending moment. Since we do not pursue any purely load alleviation driven designs but load-neutral power increase designs, the objective function is based only on the maximization of power when the loads
are neutral or negative compared to the baseline. When loads are higher than 2% (an empirical limit accounting for model uncertainty), the objective has a 90% weight on loads and 10% on power. A smooth Gaussian filter is used for the transition between neutral and higher loads (Fig. 5).



## 3.1 Surrogate modelling

For generating the initial sample set, MATLAB's Latin Hypercube design is used, with the maximin option and 20 iterations.
The minimum sample size is used which is $3*d+1$, where $d$ is the number of design variables, in our case 11. In the initial set, a reduced cubic polynomial regression model is fitted. The choice of the surrogate model is decided based on prior studies of accuracy, comparing it with quadratic regression polynomials and radial basis functions.

## 3.2 Optimization

Using the fitted surrogate model on the initial set, a global optimization approach is followed, utilizing MATLAB's genetic
algorithm with default settings. The best performing design point is chosen for a HAWC2 evaluation, together with points created by randomly perturbing the best point found so far. Also a set of points that is uniformly selected from the whole variable domain is generated (using again a Latin Hypercube design) and the score is calculated over both sets of points. Hence, it is possible to improve the global fit of the surrogate model and new areas of the variable domain where the global optimum may be located can be detected. Based on the available number of CPUs, 20 iterations are chosen, resulting in a total
number of 174 HAWC2 evaluations, including the initial sample set.

## 4   Results

The progress of the optimization and the results for the whole set of evaluated design samples is discussed here. The characteristics of the best converged design are also discussed in detail.

## 4.1   Optimization results

The progress plot showing the best value of the objective function during the evaluation of each sample is shown in Fig 6. It is seen that the objective function value is improved considerably from the starting samples and practically converges after 140 evaluations. All evaluated samples are plotted in Fig 7 in the state space of the two metrics, generator power and ultimate blade root flapwise bending moment, coloured by the value of the objective function (weighted sum). A Pareto front is clearly defined, with the best points laying on the front close to the zero load difference level. The optimal design point is designated
by the red circle.

The best design in terms of the minimum value of the objective function comprises a tip extension with a length close to the limit of the defined length (7%), with all 11 design variables listed in Table 2. The design is shown to be a slender, backwards swept and highly upwind prebent tip shape, with fore positioning of the shear center, lower rotor speed and higher pitch settings. In Fig. 8, the 3D geometry of the baseline and optimized blade tip is compared.

The blade centerline of the optimized design is compared to the baseline in Fig. 9, where the sweep and prebend offsets are shown. The optimized planform is compared to the baseline in Fig 10. The mass and flapwise stiffness distributions are compared to the baseline in Fig. 11. The optimized distributions are generally smooth and realizable, with the exception of the

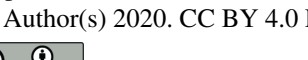



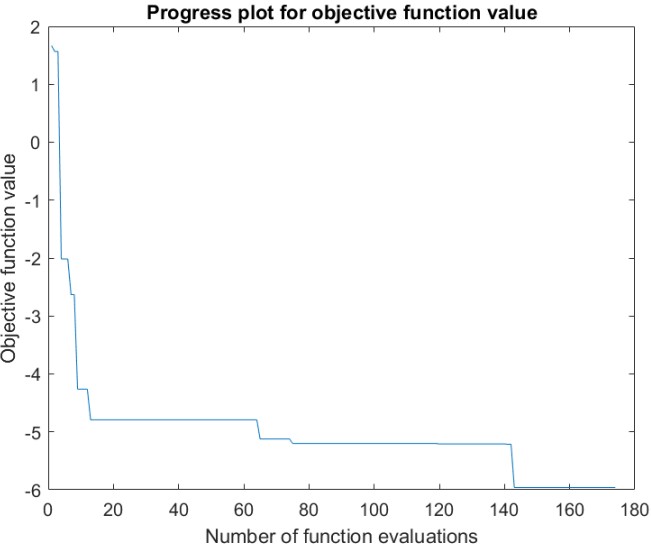

**Figure 6.** Progress plot of the objective function value.

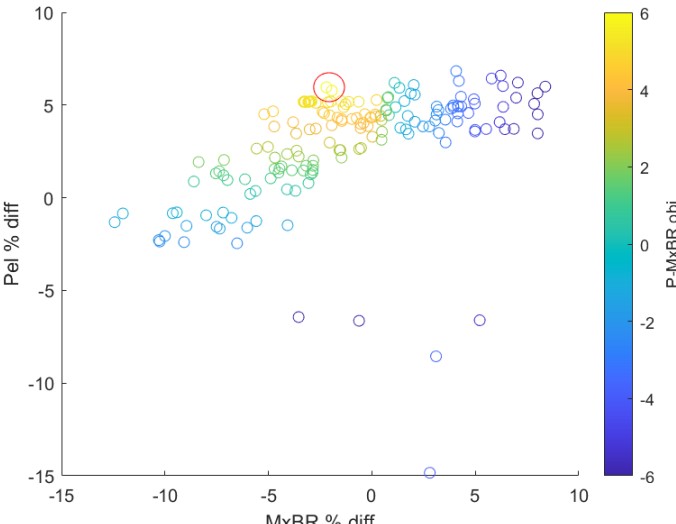

**Figure 7.** Pareto front of the evaluated samples.

twist, which in a realistic application would have to be a smooth continuation of the maximum value of the baseline, inboard of the tip.



**Table 2.** Optimized design variables.

| variable | length [%] | chord 1 [%] | chord 2 [%] | twist 1 [deg] | twist 2 [deg] | dihedral [deg] | sweep [deg] | sc off [%] | E/G scaling [%] | K opt scaling [%] | pitch opt off [deg] |
|---|---|---|---|---|---|---|---|---|---|---|---|
| value | 6.98 | 44.93 | 73.59 | 3.70 | 4.64 | -23.52 | 7.15 | 19.21 | 94.18 | 122.67 | 1 |

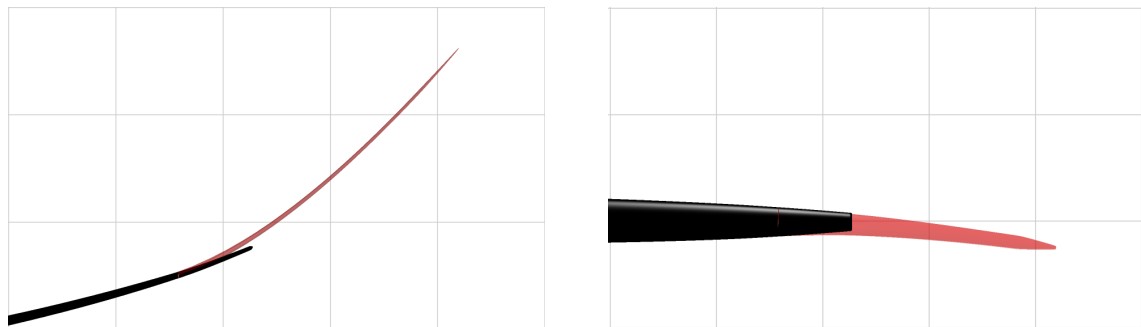

**Figure 8.** Blade 3D surface comparison between baseline geometry (in black) and optimized tip extension (in red). Left: in-plane view, right: out-of-plane view. For reference, a background grid with a spacing of 3.5 m is included.

## 4.2 Evaluation of optimized design

The performance of the optimized design is evaluated in terms of AEP in its IEC wind class I, and the lowest average wind speed class III. The 'clean' power curve is defined by steady uniform wind speed inflow from cut-in to cut-out with 1m/s steps, with no wind shear or turbulence. The higher fidelity aerodynamic module MIRAS is also used to run the same cases for comparison. The results are shown in Table 3, with the power curves plotted in Fig. 12. We see that MIRAS over-predicts the AEP for the baseline with around 1% deviation, and under-predicts the increased AEP due to the tip by around 1-2%.

The performance of the optimized design is also evaluated in the DLC1.3 (ETM) case which is used in the optimization, performed with the NW method against two different fidelity models. The BEM model implemented in HAWC2 is used as the lower fidelity method and the lifting line aerodynamic module implemented in MIRAS, which is employed as the higher fidelity solver. To ensure a meaningful comparison between the solvers, simulations of the extreme turbulent case have been carried out as follows: Firstly, to run a free turbulent simulation in MIRAS, the velocity defined Mann turbulent box used in the optimization procedure has been transformed into a particle cloud by computing the curl of the velocity field. This cloud is slowly released one diameter upstream the turbine and it develops as it convects downstream towards the rotor plane. The released turbulent particles interact freely with the turbine wake. Vortex simulations with and without the turbine are performed. In the simulation without a turbine, the local velocities are extracted every time step at the rotor plane position, in a 64x64 mesh with a cell size of approximately 6.25m. These velocities differ from the initially defined turbulent box due to the downstream development of the flow in MIRAS. Such velocities are used to generate a new turbulent field which will be loaded in the HAWC2-NW and HAWC2-BEM simulations. Such turbulent field will mimic the turbulence seen by the turbine in MIRAS,





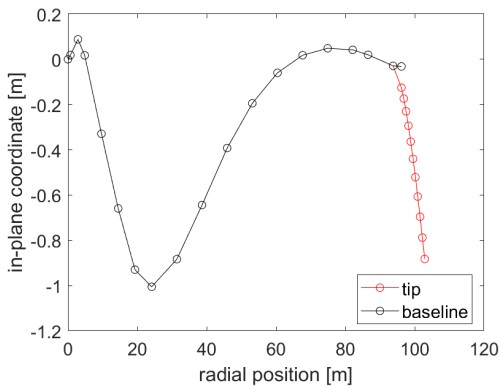 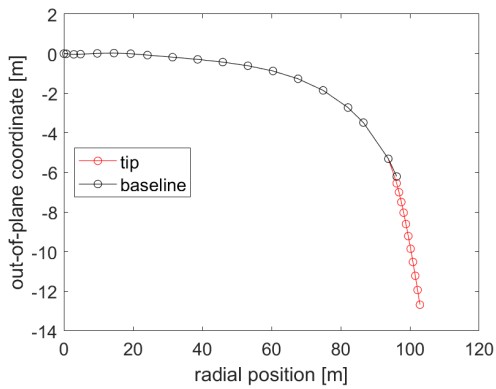

**Figure 9.** Blade centerline of the optimized design.

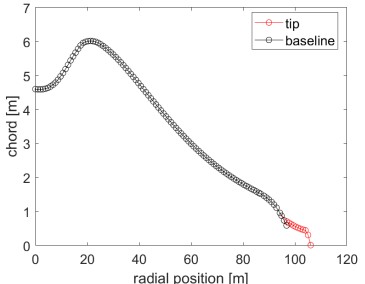 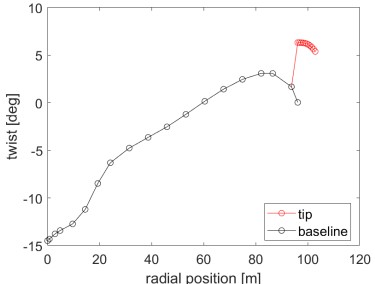 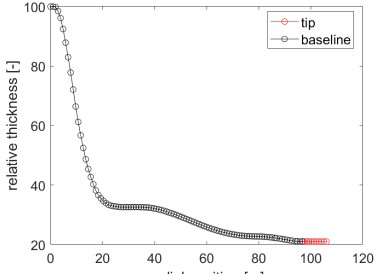

**Figure 10.** Planform of the optimized design.

although the presence of the turbine and its wake will modify the turbulent field and such phenomena can not be accounted for. In order to reduce uncertainties related to the turbine control in HAWC2, both the azimuthal rotor position as well as the pitch
angle for each one of the blades are forced to be the same as the ones computed in the MIRAS simulation (with turbine), as shown in Figures 13. In order to have a smooth start of the HAWC2-BEM and HAWC2-NW simulations, the first 60 seconds of the rotational speed signal from MIRAS have been modified using a hyperbolic tangent function. Differences between the simulations are therefore mainly related to the wake and flow modelling. The visible rotor speed differences (left plot of Figure 13) appear between the baseline rotor and the rotor with optimized tip, and the different fidelity levels clearly operate at the
same rotor speeds from 100 seconds simulated time. The pitch angles also agree well between fidelity levels. The main offset between the baseline and extended rotors is the minimum pitch angle, that is reduced to -1 degree by the optimization routine, see Table 2.

General statistics of the aerodynamic power and the flap-wise root bending moment are presented in Table 4. Regarding the power, it seems like the BEM method is slightly closer to the LL predictions in both mean and standard deviations for both
rotor designs. However, regarding the MxBR it is for all quantities except of the min predicted value of the optimized blade that the NW model is closer to the LL calculations. This is specially remarkable when looking at the standard deviation, where





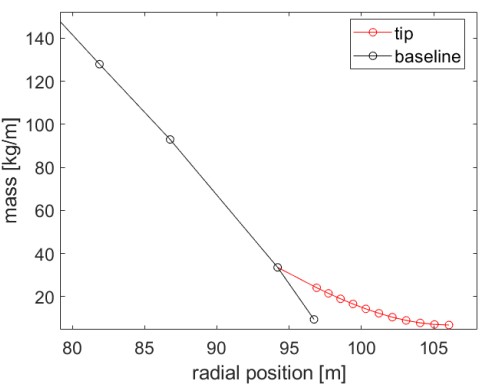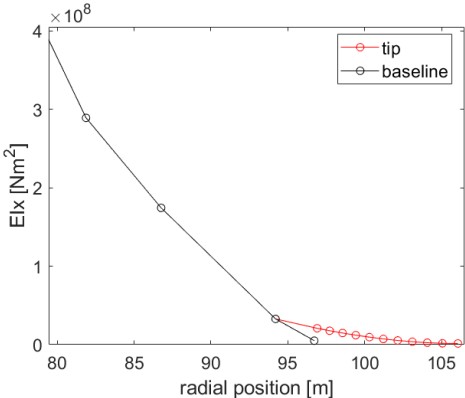

**Figure 11.** Mass and flapwise stiffness of the optimized design (zoom in tip).

**Table 3.** Comparison of AEP predictions for the baseline and optimized designs between HAWC2-NW and HAWC2-MIRAS.

| case | AEP - class I [kWh] | AEP - class III [kWh] | AEP diff - class I [%] | AEP diff - class III [%] |
|---|---|---|---|---|
| baseline - NW | 5.22e10 | 3.75e10 | - | - |
| baseline - MIRAS | 5.26e10 | 3.79e10 | 0.73 (*) | 1.10 (*) |
| opt tip - NW | 5.41e10 | 3.97e10 | 3.62 (*) | 5.96 (*) |
| opt tip - MIRAS | 5.38e10 | 3.93e10 | 2.23 (+) | 3.76 (+) |

(*): relative to baseline - NW

(+): relative to baseline - MIRAS

the NW deviation from the LL simulations is $50\%$ smaller than BEM. Note here that the defined DLC 1.3 case has a turbulent intensity level of $40\%$.

**Table 4.** Statistics of the power and flap-wise root bending moment predicted in the extreme turbulence case by HAWC2-NW and HAWC2-BEM respect to HAWC2-LL predictions. The table shows the difference in per cent.

| | BASELINE | | | OPT_TIP | | |
|---|---|---|---|---|---|---|
| | LL | NW | BEM | LL | NW | BEM |
| **mean(Power)** | - | -13.13 | -12.88 | - | -13.11 | -12.22 |
| **std(Power)** | - | -9.54 | -8.66 | - | -11.36 | -9.75 |
| **mean(MxBR)** | - | -5.83 | -6.81 | - | -5.41 | -6.04 |
| **min(MxBR)** | - | 1.22 | 2.42 | - | 2.01 | 3.54 |
| **std(MxBR)** | - | 5.08 | 6.97 | - | 5.49 | 7.56 |





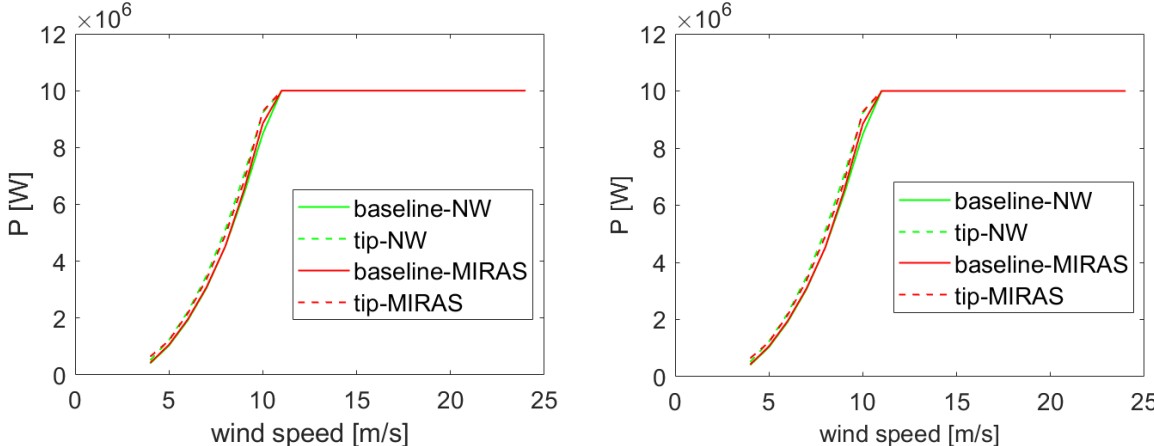

**Figure 12.** Power curve comparison between HAWC2-NW and HAWC2-MIRAS - Left: class I, Right: class III. Solid lines represent the baseline blade while dashed lines represent the optimized tip design.

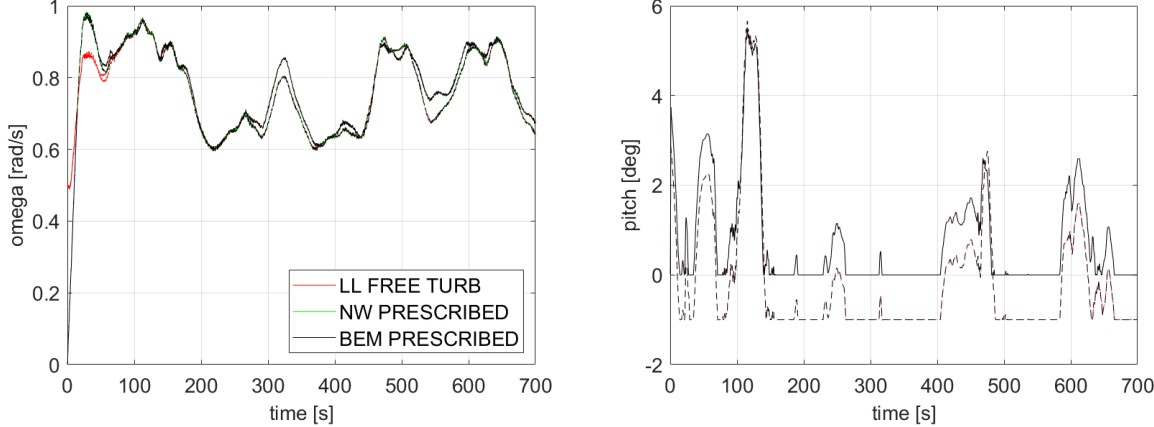

**Figure 13.** (left) rotational speed and (right) pitch signal of the baseline and optimized design. Solid lines represent the baseline blade while dashed lines represent the optimized tip design.

In order to study the influence of the turbulence level in the results, the extreme turbulence level of the DLC 1.3 (40%) has been down-scaled to obtain inflow fields with a range of turbulence intensities from 0 to 40%. Statistics of the difference in the BEM and NW predictions respect to the LL simulations in function of the turbulence level are presented in what follows. Figures 14 depict the mean and standard deviation of the power signal. In terms of the mean, there is a clear increase in the differences respect to the LL simulations with the increasing turbulent intensity. The standard-deviation of the power signal follow a different pattern, with the smallest differences between the codes obtained for a TI of 20%. An analysis of the mean, minimum and standard-deviation of the flap-wise root bending moment signal is presented in 15. In this case, differences in the standard deviation of the signal are small for the optimized tip at low TI and larger for the baseline, as the TI increases





the differences increase and align for both rotors. A similar picture is observed in the behaviour of the minimum root bending moment values, although generally differences between the rotors increase with the TI. In terms of the mean values, differences respect the LL predictions grow with increasing TI, with the engineering models predicting a higher moment at low TI and a
lower one at high TI.

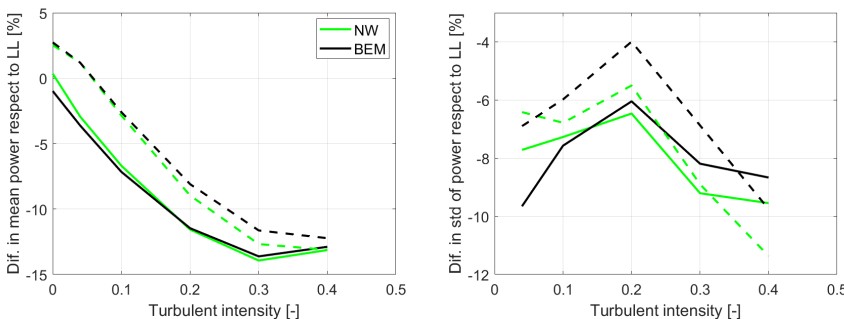

**Figure 14.** Differences in power in function of turbulence intensity (left) mean (right) standard deviation. Solid lines represent the baseline blade while dashed lines represent the optimized tip design.

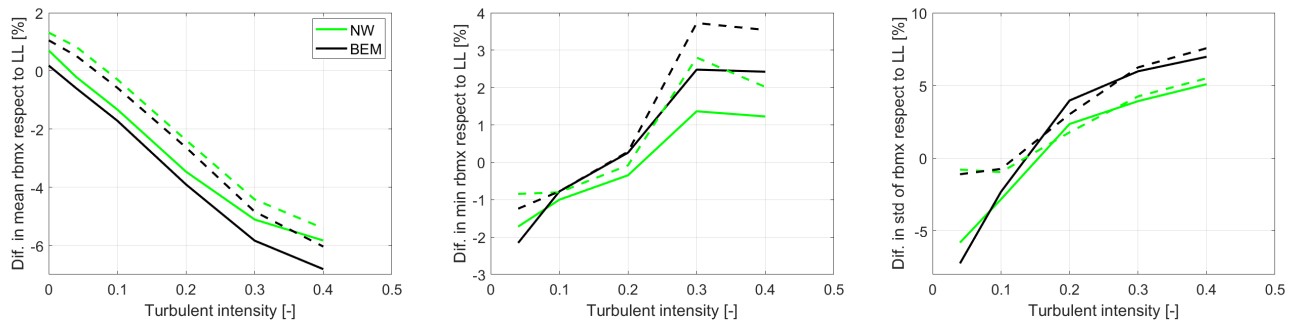

**Figure 15.** Differences in flap-wise root bending moment in function of turbulence intensity (left) mean (center) minimum (right) standard deviation. Solid lines represent the baseline blade while dashed lines represent the optimized tip design.

A detail analysis is carried out for the 40% turbulent case. Figure 16 shows the time signal of the aerodynamic power for both blade designs with the three different fidelity models. Generally there is a good agreement between the three solutions. However, there is a small power offset, where the LL predictions are most of the time slightly larger than the BEM and NW predictions.
The time variation of the predicted flap-wise root bending moment is shown in Figure 17. Opposite to what is observed for the power signal, there is no clear offset between the LL and the NW/BEM model predictions. It is visible that the BEM calculations experience larger high frequency variations compared to the higher fidelity models, exhibiting a larger standard deviation as it has previously been shown.





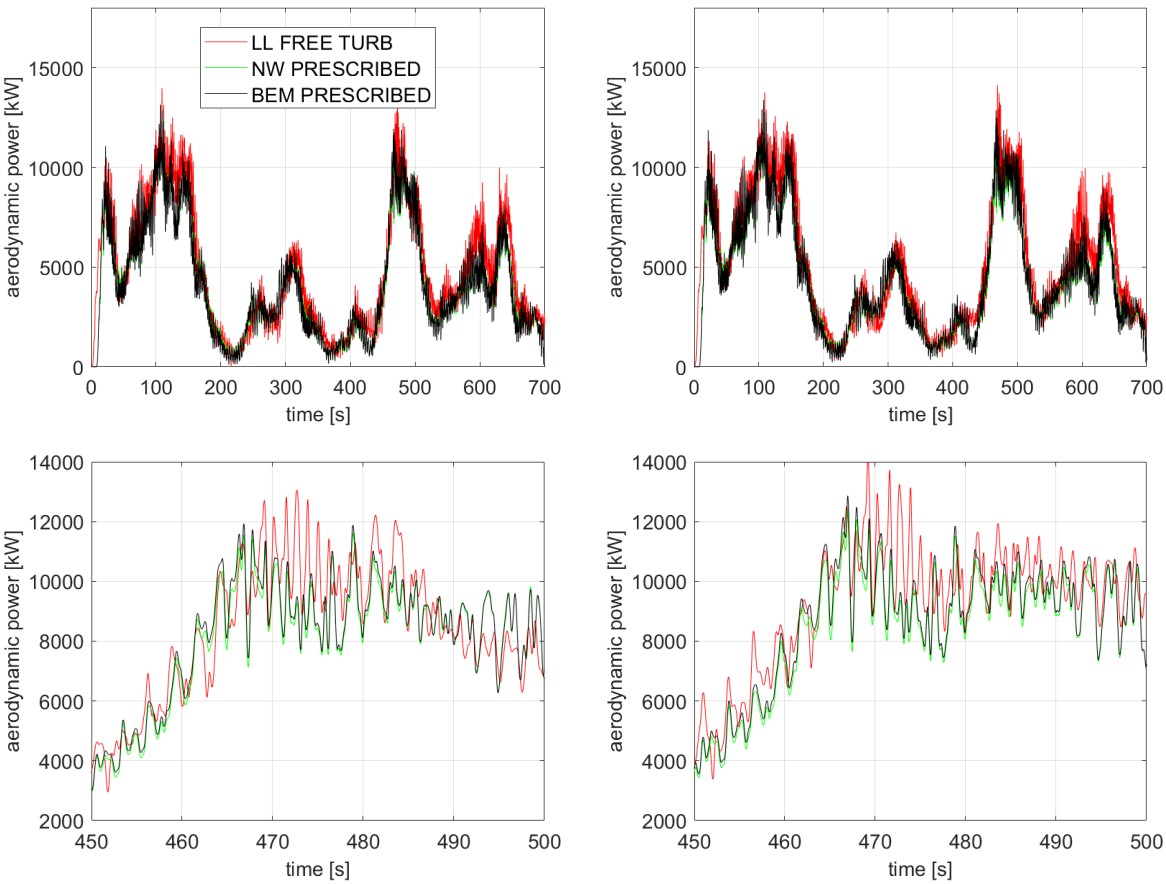

**Figure 16.** Aerodynamic power signal (left) baseline (right) optimized design. (top) full simulation time (bottom) zoom-in for a 50 seconds time interval.

The radial distributions of mean and standard deviation of AOA for the baseline blade (solid lines) and extended blade (dashed lines) are shown in Figure 18. It can be seen that the mean AOA on the tip increases by roughly 4 degrees at the start of the tip extension, which is mainly due to the twist distribution, see Figure 10. The MIRAS and the NW results both show decreased mean AOA inboard of the tip and an increased AOA on the tip itself. This is likely due to a load redistribution due to two factors: the offset of the trailed vorticity at the very blade tip and the velocity induced by the curved bound vorticity on the tip extension, Li (2018). In the standard deviation of the AOA (right plot of Figure 18) it can be seen that all codes predict the standard deviation to increase on the tip extension compared to the outboard part of the baseline blade. This is partly because the sweep angle reduces the fraction of the relative velocity that is in the planes of the aerodynamic sections, while it does not affect the wind speed changes due to turbulence. Thus the same turbulence will lead to larger variations in AOA on the swept extended tip part of the blade than on the straight outboard part of the original blade. It can also be seen in all fidelity levels that the AOA is varying less on the section between 80 and 95 meter radius on the extended blade than on the original blade,





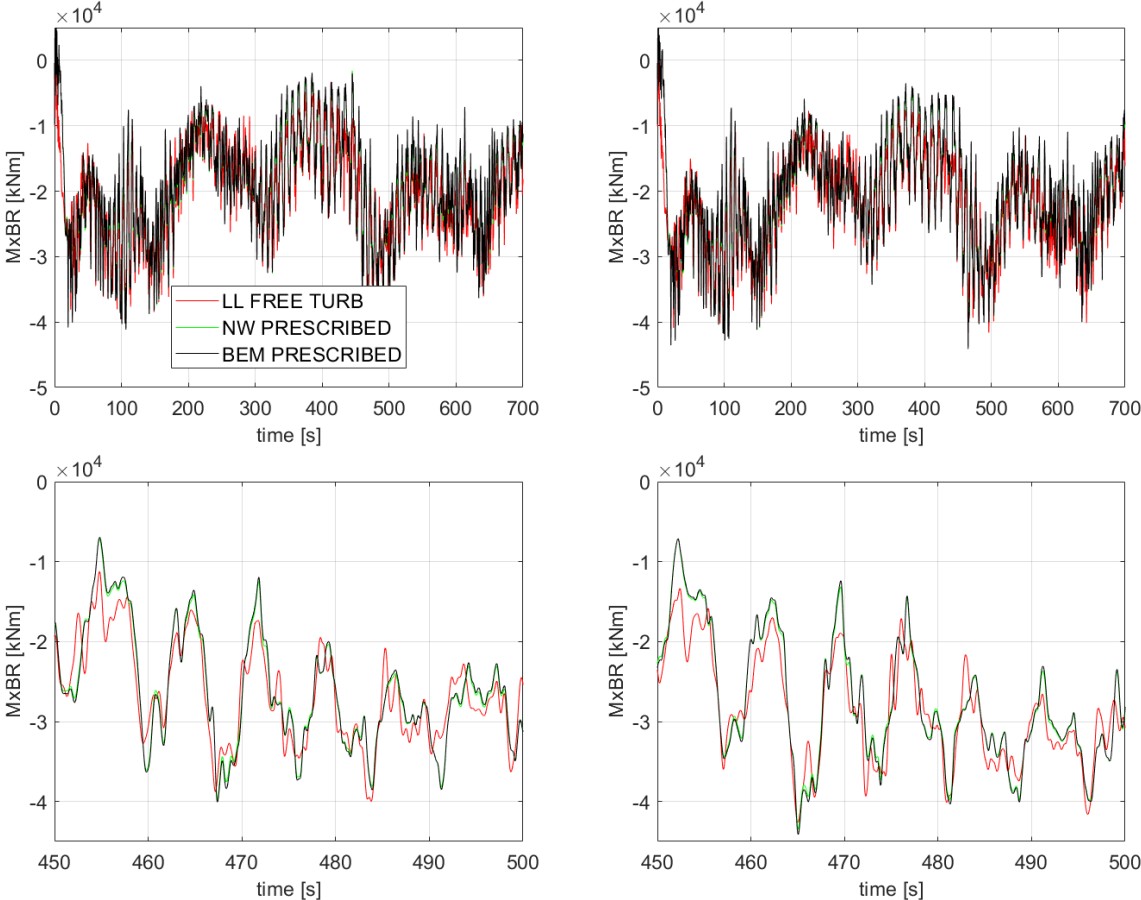

**Figure 17.** Flap-wise root bending moment, MxBR (left) baseline (right) optimized design. (top) full simulation time (bottom) zoom-in for a 50 seconds time interval.

which is due to the aeroelastic load alleviation effect that the swept tip provides. The higher mean and standard deviation of the AOA inboard of 80 meters radius on the extended blade is due to the reduced minimum pitch angle and due to the slightly reduced rotor speed, see Figure 13.

The mean and standard deviation of the in-plane force are shown in Figure 19. The overprediction of the mean in-plane force in the LL simulations corresponds to the mean power overprediction shown in Table 4. Again the load redistribution on

the swept tip can be seen clearly, with increasing loads on the tip itself and decreasing loads inboard of the tip predicted by the LL and NW models compared to the BEM code prediction. All codes predict very similar standard deviations of the in-plane force.

Similar effects can be seen in the out-of-plane forces in Figure 20. The standard deviation of the NW computations is consistently below the BEM predictions, which is in good agreement to the comparisons in (Madsen, 2018). The aeroelastic

load alleviation due to the geometric bend-twist coupling caused by the swept tip is clearly visible inboard of the tip section




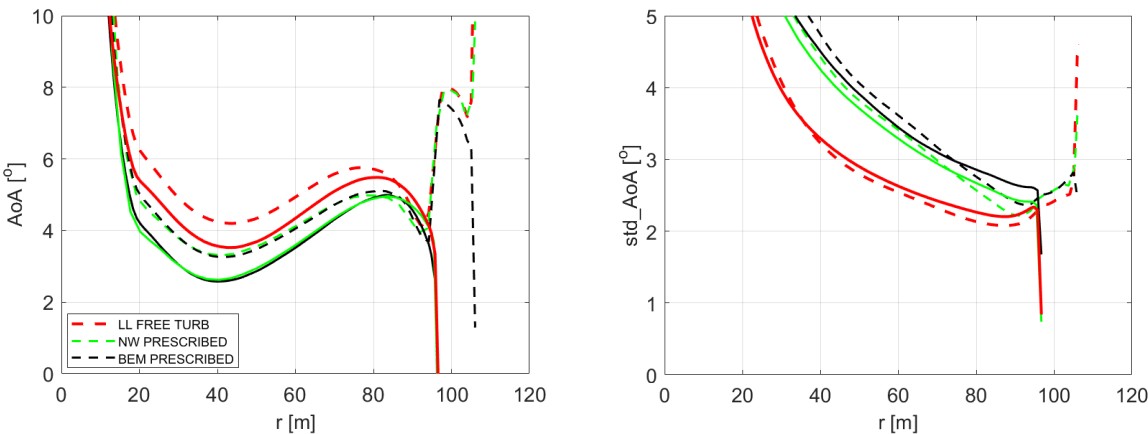

**Figure 18.** Angle of attack (left) mean distribution (right) standard deviation. Solid lines represent the baseline blade while dashed lines represent the optimized tip design.

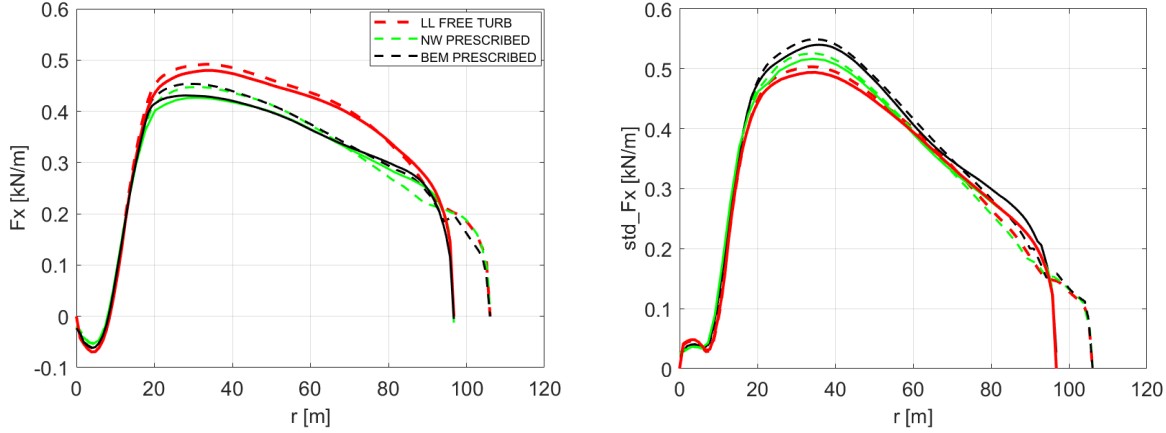

**Figure 19.** In-plane force (left) mean distribution (right) standard deviation. Solid lines represent the baseline blade while dashed lines represent the optimized tip design.

down to a radius of 50 meters. In that part of the blade, all three codes predict lower load variations for the extended blade than the baseline blade. The fact that the BEM computations agree with the higher fidelity codes very well on this load reduction indicates that it is an aeroelastic effect.

## 5 Conclusions

A novel surrogate based optimization framework for aeroelastic design of tip extensions on modern wind turbines is presented in this work. The design of tip extensions is performed in a realistic design space and aeroelastic operation of the wind turbine and it is highly efficient in terms of use of computational resources. The optimized design achieving load neutral 6% AEP



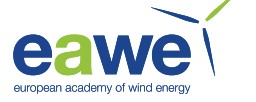


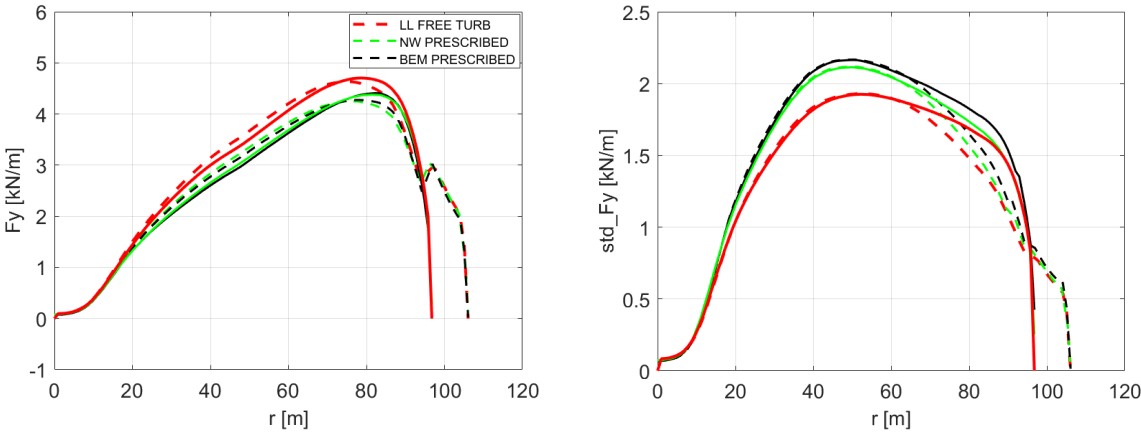

**Figure 20.** Out of plane force (left) mean distribution (right) standard deviation. Solid lines represent the baseline blade while dashed lines represent the optimized tip design.

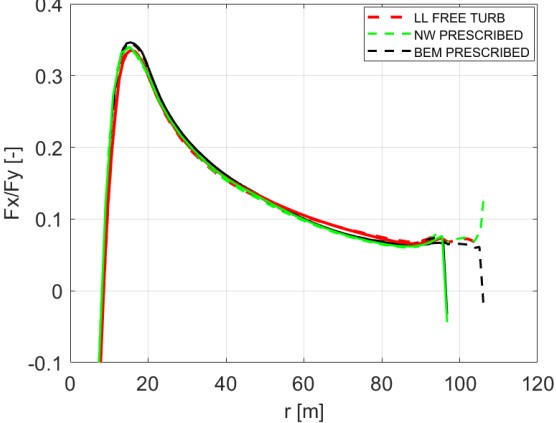

**Figure 21.** Mean values of the aerodynamic efficiency, Fx/Fy. Solid lines represent the baseline blade while dashed lines represent the optimized tip design.

gain is evaluated in detail with two levels of aerodynamic model fidelity. The aeroelastic response predictions of the complex tip shape with the Near Wake aerodynamic module agree fairly well with the higher fidelity MIRAS simulations. Detailed
comparison including a BEM model shows that local load distributions are predicted better by the near wake model, but the improvement in terms of mean power and blade root loading over the BEM model is not clear. This indicates that the coupling factor computation in the near wake model should be revisited. The agreement between the lower fidelity BEM model, the near wake model and the higher fidelity MIRAS model worsens with increasing turbulence intensity, which should be investigated in more detail in future work. The tip extension design concept resulting from the SBO process has high potential in terms
of actual implementation in a real rotor upscaling with a potential business case in reducing the LCOE of future large wind



turbine rotors. Future work will focus on introducing multi-fidelity optimization methods, but also concept innovations which could further increase the achieved performance potential.

*Code and data availability.*  The SBO framework basic Matlab code is freely available (Müller, 2014). Pre/post-processing scripts and data sets available upon request. The aeroelastic code HAWC2 is available with a license.

*Author contributions.*  Thanasis Barlas performed the aeroelastic model setup, optimization framework setup, design parametrization, design optimization, and design evaluation. Néstor Ramos García performed the MIRAS simulations and contributed to the design evaluation. Georg Pirrung contributed to the aeroelastic model setup, design parametrization, and design evaluation. Sergio González Horcas contributed to the design parametrization and design evaluation.

*Competing interests.*  No competing interests are present.

*Acknowledgements.*  This research was supported by the project Smart Tip (Innovation Fund Denmark 7046-00023B), in which DTU Wind Energy and Siemens Gamesa Renewable Energy explore optimized tip designs. The following persons have also contributed to the presented work: Helge A. Madsen, Flemming Rasmussen, Niels N. Sørensen, Peder B. Enevoldsen, Jesper M. Laursen.



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
