# Peer review of "Surrogate based aeroelastic design optimization of tip extensions on a modern 10MW wind turbine"

_Wind Energy Science, 2020_

## Referee Comment (RC1) · Niels Adema (Referee) · 18 Nov 2020

The article being reviewed details a novel tip design and optimization approach for wind turbine blades. It outlines in great detail an optimization framework for blade tip extension based on surrogate aeroelastic modelling. Tip extensions are parameterized using 11 variables on 10 additional structural and aerodynamic sections. An academic wind turbine is modelled in an aeroelastic design code and simulations are done using a near wake module. A surrogate model is fitted to data from full aeroelastic simulations in extreme turbulent conditions. The tip optimization method is seeking to increase power production while maintaining neutral loading. To achieve this the

objective function op the optimization is a weighted sum of the generator power and the ultimate blade root flapwise bending moment. The optimization results are evaluated for AEP in both IEC wind class I and III. Further performance is evaluated using DLC1.3 for higher turbulence levels using the near wake model that is used in creating the surrogate model. And furthermore, it is compared with two different fidelity models namely: a BEM model implemented in HAWC2 (lower fidelity) and a lifiting line method implemented in MIRAS (higher fidelity). From the resulting figures and explanation of the results it becomes clear that for lower turbulence the solvers show good agreement between blade root bending moments and generator power. With increasing turbulence the agreement between solvers shows greater difference, which should be considered in future work. While the higher turbulence results require more research the overall results show high potential for this tip extension design process. From the referees point of view the article is sufficiently well written for reproduction of the study by other scientists. This study will add significantly to the implementation of rotor up-scaling and the application of novel tip design in large rotor concepts. The increase in AEP of up to 6% will add to lower LCOE of future wind turbines.

Considering the above I recommend publishing of this study taking into consideration the question below and a few (very) minor textual and formatting comments posted below.

Question: In line 142 to 144 it is mentioned that, besides the new twist distribution, the new tip design is realistic for application in new blades. The optimized results for the twist distribution do not show a smooth continuation of the maximum values inboard of the baseline tip. The referee would find it interesting to see the impact on optimization results, the corresponding aerodynamic performance, and blade loading when considering a realistic twist distribution in the optimization of a new tip design. This could be part of future work which might have large implications in a new tip design.

Textual and formatting comments:

- Consider reformatting such that the following lines are attached to their paragraph and not in between or after figures on another page: lines 143&144, 177&178

- Table 1. Consider adding line explaining that min values are for Tip 2 in Figure 2 and max values are for Tip 1. This is not mentioned and makes the legend of figure 2 slightly confusing.

- Line 110, consider rephrasing to "When the increase in loads is higher than 2%....."

- Line 115, rephrase to "The minimum sample size used is 3*d+1, where......"

- Line 133, consider rephrasing to "A Pareto front is clearly visible, with....."

- Line 171, Minimum pitch angle is reduced to -1 degrees. However Table 2 shows opt Pitch off = 1 degrees. Is this a typo?

- Caption table 4, change per cent to percentages.

- Line 185, add "figure" before 15 such that: "..... moment signal is presented in figure 15".

- Line 191, rephrase to "A detailed analysis......"

---

## Short Comment (SC1) · 4 Jan 2021

The manuscript describes a methodology how to optimize a parametrized blade tip extension planform with respect to maximizing Levelized Cost of Energy (LCoE), i.e., minimizing flap-wise blade root bending moment while maximizing the annual energy production. Moreover, a high and a low fidelity simulation are compared. The work concludes some interesting shape tendencies of the optimum blade tip extension planform.

In their literature review, the authors claim that they were the first to investigate the blade tip extension with focus on performance and loads by means of a "real operational case with a view toward a business case" [page 1, line 20ff]. At least, Rosemeier et al. (2020), however, used a full set of "real" IEC design load cases to assess the feasibility of a blade extension. Moreover, their study focused on a clear and relevant business case, i.e., the supplement of a blade tip extension to a lifetime extension scenarios.

I suggest to define more clearly the novelty of your research to appropriately contrast to the existing literature.

---

## Short Comment (SC2) · 4 Jan 2021

The authors appreciate the feedback in the short comment. After the official reviewes' comments have been received, we will revise the text on the article novelty compared to the available literature in order to accurately reflect the contribution of Rosemeier et al. (2020).

---

## Referee Comment (RC2) · Rad Haghi (Referee) · 17 Jan 2021

The authors try to use a surrogate model to optimize a DTU 10MW turbine's tip extension. The topic of tip extension is a fascinating topic to address. The followings are my comments and concerns for this piece of work.

Reading the title, I think it is a bit misleading. I expected the paper to have a more elaborate explanation about the used optimization method and the surrogate model part. The authors look at the surrogate model and optimization and black boxes from what I understood. There is not that much about the assumptions and methods used for the optimization part.

It seems the optimization is done only for one wind speed (8m/s) for DLC 1.3. You expect the simulation to reach the maximum load in the simulation's length. What is your simulation length? How are you sure that a specific turbulence seed provides you with the max load?

In the top extension parametrization section, you mention, "Their range is a result of many prior parametric studies, and it is limited to ensure the validity of the aerodynamic modeling." (L62) I think this needs a reference. In the same section, "The distribution of the planform variables is calculated with a cubic Hermite interpolating polynomial." Why? I think it worth an explanation.

The part about the surrogate model is way too short. It is unclear about your assumptions and reasoning for the surrogate model part. Why did you used Latin Hypercube and not another sampling method? What type of surrogate model are you using (PCE, Kriging, etc.)? It seems you used polynomials; if that is the case, what is the reasoning behind that? Why do you use only 34 samples? That seems not enough to fit a surrogate model on your results. You refer to the previous studies (L116); what are they?

L 121: "Also a set of points that is uniformly selected from the whole variable domain is generated (using again a Latin Hypercube design) and the score is calculated over both sets of points." This sentence is unclear. Please re-evaluate.

L 124: Your statement about 20 iterations and a total number of 174 HAWC evolutions is puzzling. I think it needs some clarification.

L 149: "We see that MIRAS over-predicts the 150 AEP for the baseline with around 1% deviation, and under-predicts the increased AEP due to the tip by around 1-2%." This sentence is unclear. Please re-write the sentence.

In Figure 12, there is no increase in the rated power. What is the reason for that? I guess the reason is your optimization works only for one pitch degree, so your results

only valuable for underrated wind speed.

The results part is extensive and well explained. The aeroelastic simulation is the authors' expertise. The results section proofs the concept. However, the simulation setup's information is limited and replicating the experiment is not possible. What was the wind speed for your simulation? How many simulations did you run? I very much like to see a comparison between the blade root moment time series of the baseline and the optimized model.

L227: Claiming LCOE reduction potential based on an increase in AEP, and load neutrality on one wind speed without considering the tip extension's effect on the turbine lifetime, seems a bit too far fetch. If there are some studies on this, I am very interested in seeing them.

General comment: It seems some of the abbreviations are not missing. Please check them out.

---

## Author Comment (AC1) · 2 Feb 2021

The authors would like to thank the reviewer for their time and greatly appreciate their feedback and suggestions to improve the article.

- L142-144: Implementing a smooth twist distribution (from a point further inboard the blade reference line) is expected to have a negligible effect on the load distribution in lifting-line type of codes. The load distributions are already smooth (Fig. 19) for NW and MIRAS where trailing vorticity gradients are taken into account. In CFD, the geometry would be by default smooth, and a good comparison with the lifting-line models has already been established:

[Figure]

Li, A., Pirrung, G., Madsen, H. A., Gaunaa, M., & Zahle, F. (2018). Fast trailed and bound vorticity modeling of swept wind turbine blades. Journal of Physics: Conference Series, 1037(6), [062012]. https://doi.org/10.1088/1742-6596/1037/6/062012

Li, A., Gaunaa, M., Pirrung, G. R., Ramos-García, N., & Horcas, S. G. (2020). The influence of the bound vortex on the aerodynamics of curved wind turbine blades. Journal of Physics: Conference Series, 1618(5), [052038]. https://doi.org/10.1088/1742-6596/1618/5/052038

- L143-144, L177-178: The placement of figures has been updated.

- Table1: Explanation text has been added to clarify the min/max values in the dotted lines in Fig1-2.

- L110,115,133: The sentences have been rephrased.

- L171: The sign of the pitch setting has been clarified in accordance with Table2.

- Table4: Caption text has been updated.

- L185: 'Fig' has been be added int the text

- L191: The sentence has been rephrased.

The changes will appear in the revised manuscript.
* * *

---

## Author Comment (AC2) · 2 Feb 2021

The authors would like to thank the reviewer for their time and greatly appreciate their feedback and suggestions to improve the article.

- Title and general content: From the authors point of view the focus of the article is not on the evaluation of the optimization method but on its application to a realistic design problem. Moreover, the authors believe that the title accurately represents the work presented in the manuscript.

- Simulation case choice: One main assumption in the choice of the single simulation

case is that it is representative of the ultimate design loads of interest. This is indeed the case for the baseline design. The 600s ETM simulation ensures that a range of inflow and operating conditions is accounted for. Moreover, different turbulence intensities are simulated in the last part of the article together with full wind speed range power curves for the AEP evaluation. The authors agree that a full DLB would be preferable, but the load simulation cases have been kept to a minimum for a fast and robust optimization setup.

- Tip parametrization: The limits of the tip geometry design space are chosen based on practical constraints (monotonically decreasing chord, realistically added tip length), model range of validity (in/out-of plane offsets), and targeted power/load capability (in/out-of-plane offsets and twist signs). The authors agree that the chosen design space limits are crucial for the optimization directions, but it would not have been practical to include all prior parametric studies in this article. Some references to prior relevant work have been added.

- Surrogate model choices: The chosen DOE and surrogate model methods were based on the cited literature connected to the utilized SBO library. The number of initial sample points in the DOE was in accordance to the model requirements and the chosen methods. The authors believe that the focus of this work is not on the evaluation of the best optimization methods but on the evaluation of the potential design benefit of the chosen methods.

- L121,124, 149: The sentences have been rephrased.

- Fig12: The benefit of such a design is to increase power extraction in below rated operation within the load constraints. The power in rated conditions is regulated by the controller to the rated power setpoint.

- Results section: The authors believe that all the information about the simulated load cases is included in the paper, so the results are easily reproducible. Following the reviewers comments, a time series comparison of the blade root moment between the

baseline and the optimized model has been added.

- L227: The authors believe that the claim for reduced LCOE of such a design concept is realistically valid. This work has been carried out in connection with industrial partners with the targeted business case being very realistic. In that sense the lifetime impact (fatigue loads) of the concept is not driving the design, although a full DLB evaluation is of course necessary for such a concept evaluation.

- Abbreviations: The missing abbreviations have been added.

The changes will appear in the revised manuscript.

---

## Author Response (AR1)

**Authors' response to review of article wes-2020-108-RC1/RC2/SC1**

**RC1**

**RC1_1:**

The article being reviewed details a novel tip design and optimization approach for wind turbine blades. It outlines in great detail an optimization framework for blade tip extension based on surrogate aeroelastic modelling. Tip extensions are parameterized using 11 variables on 10 additional structural and aerodynamic sections. An academic wind turbine is modelled in an aeroelastic design code and simulations are done using a near wake module. A surrogate model is fitted to data from full aeroelastic simulations in extreme turbulent conditions. The tip optimization method is seeking to increase power production while maintaining neutral loading. To achieve this the objective function op the optimization is a weighted sum of the generator power and the ultimate blade root flapwise bending moment. The optimization results are evaluated for AEP in both IEC wind class I and III. Further performance is evaluated using DLC1.3 for higher turbulence levels using the near wake model that is used in creating the surrogate model. And furthermore, it is compared with two different fidelity models namely: a BEM model implemented in HAWC2 (lower fidelity) and a lifting line method implemented in MIRAS (higher fidelity). From the resulting figures and explanation of the results it becomes clear that for lower turbulence the solvers show good agreement between blade root bending moments and generator power. With increasing turbulence the agreement between solvers shows greater difference, which should be considered in future work. While the higher turbulence results require more research the overall results show high potential for this tip extension design process. From the referees point of view the article is sufficiently well written for reproduction of the study by other scientists. This study will add significantly to the implementation of rotor up-scaling and the application of novel tip design in large rotor concepts. The increase in AEP of up to 6% will add to lower LCOE of future wind turbines.

**AC1_1:**

The authors would like to thank the reviewer for their time and greatly appreciate their feedback and suggestions to improve the article.

**RC1_2:**

In line 142 to 144 it is mentioned that, besides the new twist distribution, the new tip design is realistic for application in new blades. The optimized results for the twist distribution do not show a smooth continuation of the maximum values inboard of the baseline tip. The referee would find it interesting to see the impact on optimization results, the corresponding aerodynamic performance, and blade loading when considering a realistic twist distribution in the optimization of a new tip design. This could be part of future work which might have large implications in a new tip design.

**AC1_2:**

Implementing a smooth twist distribution (from a point further inboard the blade reference line) is expected to have a negligible effect on the load distribution in lifting-line type of codes. The load

distributions are already smooth (Fig. 19) for NW and MIRAS where trailing vorticity gradients are taken into account. In CFD, the geometry would be by default smooth, and a good comparison with the lifting-line models has already been established:

Li, A., Pirrung, G., Madsen, H. A., Gaunaa, M., & Zahle, F. (2018). Fast trailed and bound vorticity modeling of swept wind turbine blades. Journal of Physics: Conference Series, 1037(6), [062012]. https://doi.org/10.1088/1742-6596/1037/6/062012

Li, A., Gaunaa, M., Pirrung, G. R., Ramos-García, N., & Horcas, S. G. (2020). The influence of the bound vortex on the aerodynamics of curved wind turbine blades. Journal of Physics: Conference Series, 1618(5), [052038]. https://doi.org/10.1088/1742- 6596/1618/5/052038

**RC1_3:**

Consider reformatting such that the following lines are attached to their paragraph and not in between or after figures on another page: lines 143&144, 177&178.

**AC1_3:**

The placement of figures has been updated.

**RC1_4:**

Table 1. Consider adding line explaining that min values are for Tip 2 in Figure 2 and max values are for Tip 1. This is not mentioned and makes the legend of figure 2 slightly confusing.

**AC1_4:**

Explanation text has been added to clarify the min/max values in the dotted lines in Fig1-2.

**RC1_5:**

Line 110, consider rephrasing to "When the increase in loads is higher than 2%.....".

Line 115, rephrase to "The minimum sample size used is 3*d+1, where......".

Line 133, consider rephrasing to "A Pareto front is clearly visible, with.....".

**AC1_5:**

L110,115,133: The sentences have been rephrased.

**RC1_6:**

Line 171, Minimum pitch angle is reduced to -1 degrees. However Table 2 shows opt Pitch off = 1 degrees. Is this a typo?

**AC1_6:**

The sign of the pitch setting has been clarified in accordance with Table2.

**RC1_7:**

Caption table 4, change per cent to percentages.

**AC1_7:**

Table4: Caption text has been updated.

**RC1_8:**

Line 185, add "figure" before 15 such that: "..... moment signal is presented in figure 15".

**AC1_8:**

L185: 'Fig' has been be added int the text.

**RC1_9:**

Line 191, rephrase to "A detailed analysis......".

**AC1_9:**

L191: The sentence has been rephrased.

**RC2**

**RC2_1:**

The authors try to use a surrogate model to optimize a DTU 10MW turbine's tip extension. The topic of tip extension is a fascinating topic to address. The followings are my comments and concerns for this piece of work.

**AC2_1:**

The authors would like to thank the reviewer for their time and greatly appreciate their feedback and suggestions to improve the article.

**RC2_2:**

Reading the title, I think it is a bit misleading. I expected the paper to have a more elaborate explanation about the used optimization method and the surrogate model part. The authors look at the surrogate model and optimization and black boxes from what I understood. There is not that much about the assumptions and methods used for the optimization part.

**AC2_2:**

Title and general content: From the authors point of view the focus of the article is not on the evaluation of the optimization method but on its application to a realistic design problem. Moreover, the authors believe that the title accurately represents the work presented in the manuscript.

**RC2_3:**

It seems the optimization is done only for one wind speed (8m/s) for DLC 1.3. You expect the simulation to reach the maximum load in the simulation's length. What is your simulation length? How are you sure that a specific turbulence seed provides you with the max load?

**AC2_3:**

One main assumption in the choice of the single simulation case is that it is representative of the ultimate design loads of interest. This is indeed the case for the baseline design. The 600s ETM simulation ensures that a range of inflow and operating conditions is accounted for. Moreover, different turbulence intensities are simulated in the last part of the article together with full wind speed range power curves for the AEP evaluation. The authors agree that a full DLB would be preferable, but the load simulation cases have been kept to a minimum for a fast and robust optimization setup.

**RC2_4:**

In the top extension parametrization section, you mention, "Their range is a result of many prior parametric studies, and it is limited to ensure the validity of the aerodynamic modeling." (L62) I think this needs a reference. In the same section, "The distribution of the planform variables is calculated with a cubic Hermite interpolating polynomial." Why? I think it worth an explanation.

**AC2_4:**

The limits of the tip geometry design space are chosen based on practical constraints (monotonically decreasing chord, realistically added tip length), model range of validity (in/out-of plane offsets), and targeted power/load capability (in/out-of-plane offsets and twist signs). The authors agree that the chosen design space limits are crucial for the optimization directions, but it would not have been practical to include all prior parametric studies in this article. Some references to prior relevant work have been added.

**RC2_5:**

The part about the surrogate model is way too short. It is unclear about your assumptions and reasoning for the surrogate model part. Why did you used Latin Hypercube and not another sampling method? What type of surrogate model are you using (PCE, Kriging, etc.)? It seems you used polynomials; if that is the case, what is the reasoning behind that? Why do you use only 34 samples? That seems not enough to fit a surrogate model on your results. You refer to the previous studies (L116); what are they?

**AC2_5:**

Surrogate model choices: The chosen DOE and surrogate model methods were based on the cited literature connected to the utilized SBO library. The number of initial sample points in the DOE was in accordance to the model requirements and the chosen methods. The authors believe that the focus of this work is not on the evaluation of the best optimization methods but on the evaluation of the potential design benefit of the chosen methods.

**RC2_6:**

L 121: "Also a set of points that is uniformly selected from the whole variable domain is generated (using again a Latin Hypercube design) and the score is calculated over both sets of points." This sentence is unclear. Please re-evaluate.

L 124: Your statement about 20 iterations and a total number of 174 HAWC evolutions is puzzling. I think it needs some clarification.

L 149: "We see that MIRAS over-predicts the 150 AEP for the baseline with around 1% deviation, and under-predicts the increased AEP due to the tip by around 1-2%." This sentence is unclear. Please re-write the sentence.

**AC2_6:**

L121,124, 149: The sentences have been rephrased.

**RC2_7:**

In Figure 12, there is no increase in the rated power. What is the reason for that? I guess the reason is your optimization works only for one pitch degree, so your results only valuable for underrated wind speed.

**AC2_7:**

Fig12: The benefit of such a design is to increase power extraction in below rated operation within the load constraints. The power in rated conditions is regulated by the controller to the rated power setpoint.

**RC2_8:**

The results part is extensive and well explained. The aeroelastic simulation is the authors' expertise. The results section proofs the concept. However, the simulation setup's information is limited and replicating the experiment is not possible. What was the wind speed for your simulation? How many simulations did you run? I very much like to see a comparison between the blade root moment time series of the baseline and the optimized model.

**AC2_8:**

- Results section: The authors believe that all the information about the simulated load cases is included in the paper, so the results are easily reproducible. A time series comparison of the blade root moment between the baseline and the optimized model is already included and discussed in Fig. 17.

**RC2_9:**

L227: Claiming LCOE reduction potential based on an increase in AEP, and load neutrality on one wind speed without considering the tip extension's effect on the turbine lifetime, seems a bit too far fetch. If there are some studies on this, I am very interested in seeing them.

**AC2_9:**

L227: The authors believe that the claim for reduced LCOE of such a design concept is realistically valid. This work has been carried out in connection with industrial partners with the targeted business case being very realistic. In that sense the lifetime impact (fatigue loads) of the concept is not driving the design, although a full DLB evaluation is of course necessary for such a concept evaluation.

**RC2_10:**

General comment: It seems some of the abbreviations are not missing. Please check them out.

**AC2_10:**

Abbreviations: The missing abbreviations have been added.

**SC1**

**SC1_1:**

The manuscript describes a methodology how to optimize a parametrized blade tip extension planform with respect to maximizing Levelized Cost of Energy (LCoE), i.e., minimizing flap-wise blade root bending moment while maximizing the annual energy production. Moreover, a high and a low fidelity simulation are compared. The work concludes some interesting shape tendencies of the optimum blade tip extension planform. In their literature review, the authors claim that they were the first to investigate the blade tip extension with focus on performance and loads by means of a "real operational case with a view toward a business case" [page 1, line 20ff]. At least, Rosemeier et al. (2020), however, used a full set of "real" IEC design load cases to assess the feasibility of a blade extension. Moreover, their study focused on a clear and relevant business case, i.e., the supplement of a blade tip extension to a lifetime extension scenarios I suggest to define more clearly the novelty of your research to appropriately contrast to the existing literature.

**ACS1_1:**

The authors appreciate the feedback in the short comment. After the official reviewes' comments have been received, we will revise the text on the article novelty compared to the available literature in order to accurately reflect the contribution of Rosemeier et al. (2020).